# Operando visualisation of lithium plating by ultrasound imaging of battery cells

**David Wasylowski** [1,2,3] ✉, **Heinrich Ditler** [1,2,3], **Morian Sonnet** [1,2,3], **Tim Falkenstein** [1,2], **Luca Leogrande** [1,2], **Emanuel Ronge** [4], **Alexander Blömeke** [1,2,3], **Andreas Würsig** [4], **Florian Ringbeck** [1,2,3] & **Dirk Uwe Sauer** [1,2,3,5,6]

While developing battery cells, the achievement of fast-charging capability is heavily dependent on avoiding metallic plating on the anode surface (i.e., lithium plating in lithium-ion cells). However, this objective hinges on the effectiveness of plating detection. Currently, measurement techniques are either inadequate in providing spatial, temporal, or causal information, incur high costs when employing, e.g., neutron imaging, or are lengthy due to destructive post-mortem examinations that additionally lack operando data. In this work, we demonstrate an ultrasound imaging method for operando visualization of the interior of a multi-layer pouch battery cell. Here we show that this method can non-invasively visualize the formation and stripping of lithium plating during cycling. Extensive reference electrode studies and ex-situ analysis verify the effectiveness of our method for plating detection. Ultimately, this work enables researchers and industry to significantly accelerate the development of new cell technologies and their optimized utilization.

The development of fast charging technologies for lithium-ion cells is widely seen as a pivotal catalyst, revolutionizing the electric vehicle (EV) landscape by mitigating range anxiety and accelerating their widespread global adoption[1]. Multiple government bodies, including the US Department of Energy, have underscored the urgency for extreme fast-charging technologies (XFC). Their mandate advocates for a fast-charging capacity of 200 miles within 10 min while ensuring minimal impact on battery aging[2]. Along with meeting the fast-charging benchmarks, a primary goal remains to attain price targets of under 80$/kWh as a crucial milestone. Achieving both objectives of fast charging and minimal costs is a challenging balancing act.

Generally, XFC can be achieved by reducing the anode coating thickness, which is accompanied by a reduction of energy density. Thus, more cells are required for the same energy at a higher overall cost[3]. Conversely, increasing the anode coating thickness increases the energy density but reduces the fast-charging capability due to the increased probability of lithium plating occurring[4]. Therefore, for the development of new battery cell technologies and their optimized use, a method to detect the deposition of metallic lithium on the anode surface in real-time during fast charging is crucial. However, operando lithium-plating detection on a multilayer-cell level has presented a significant challenge for the scientific community in recent years. A wide variety of operando methods have been developed that can generally be classified into four categories: Coulombic efficiency and voltage-based approaches[5–7], impedance measurements[8–10], electro-chemical modeling[11,12], and mechanical measurements[13–15]. All these methods have demonstrated that under specific conditions and considering certain model assumptions, indirect detection of lithium

[1]Chair for Electrochemical Energy Conversion and Storage Systems, Institute for Power Electronics and Electrical Drives (ISEA), RWTH Aachen University, Aachen, Germany. [2]Center for Ageing, Reliability and Lifetime Prediction of Electrochemical and Power Electronic Systems (CARL), RWTH Aachen University, Aachen, Germany. [3]Jülich Aachen Research Alliance, JARA-Energy, Aachen, Germany. [4]Battery Systems FAB-SH, Fraunhofer Institute for Silicon Technology, Itzehoe, Germany. [5]Institute for Power Generation and Storage Systems (PGS), E.ON ERC, RWTH Aachen University, Aachen, Germany. [6]Helmholtz Institute Münster (HI MS), IEK 12, Forschungszentrum Jülich, Jülich, Germany. ✉e-mail: david.wasylowski@isea.rwth-aachen.de

plating is possible. However, a central drawback of these operando methods is their provision of one-dimensional information. This limitation makes it challenging to measure local electrochemical effects, especially in large cells used in applications relevant to the energy transition[16,17]. Additionally, electrochemical methods, particularly in large cells, suffer from uncertainty in temperature distribution due to strong temperature dependence as well as uncertainties in the model parameters[18,19]. In recent literature, neutron imaging has also been used to visualize lithium plating, which, apart from the generation of nuclear waste, along with safety and regulatory concerns, is associated with very high costs, low accessibility, and low parallelizability, which severely limits the usefulness of the method for rapid development of new cell technologies[20]. Consequently, despite numerous studies, there hasn't been a viable and real-time method available for providing spatial, temporal, and geometric information regarding lithium plating formation[17,21].

In this work, we bridge this gap by presenting a method for achieving operando and non-destructive visualization of the internal dynamics within a battery cell using Scanning Acoustic Microscopy (SAM). Contrary to the pioneering transmission-based work by Chang et al. and Robinson et al., this method is based on the emission and reflection of ultrasound waves, which are subsequently analyzed using our signal processing toolchain based on Wasylowski et al.[22–24]. By directly imaging the cell's electrodes, the effectiveness of this method remains independent of the cell's aging, cell-to-cell variability, and temperature (distribution). Furthermore, minimal parameterization of the measurement process is required in contrast to methods based on electrochemical modeling. With our setup, we show the successful 2D visualization of lithium plating in multilayer pouch cells without relying on neutron-based methods. This method enables a significantly easier and more cost-effective approach to visualizing battery cells in real-time during the development phase, as well as to designing fast-charging protocols.

## Results
### Setup and imaging
The SAM method generates an ultrasound wave at every position (i.e., pixel) of the battery cell with a resolution of 75 µm. For this purpose, a piezoelectric ultrasound transducer with a resonance frequency of 25 MHz is electrically stimulated by a pulse generator, causing it to emit a sound wave with the same center frequency into space. This wave then travels through a liquid (here distilled water) with a significantly lower damping coefficient and much greater acoustic impedance ($Z$) compared to air until it reaches the cell surface. At the cell surface, the ultrasound wave is partially reflected and transmitted. The transmitted portion continues to travel through the electrode stack until it encounters another material interface, where it undergoes further transmission and reflection. The reflection and transmission fractions can be calculated using Eqs. (1) and (2) from chapter 4. Further information can be found in chapter 4 and in ref. 25. To enable visualization in the form of images, the sensor must be moved along the entire cell surface. Focused sensors, which concentrate ultrasound energy at a defined distance, calibrated and specified by the manufacturer, were used for these measurements. This allows most of the energy of the ultrasound wave to be focused within the cell, minimizing the reflection at the cell surface to achieve the best possible signal-to-noise ratio at the point of interest. This principle is illustrated in Fig. 1a. As explained in chapter 4, due to the significant differences in acoustic impedances between water and pouch bag material, the largest reflection still occurs at the cell surface but can be ignored due to the focal point being inside the electrode stack.

For operando lithium plating visualization, a verified method for inducing lithium plating was first chosen and applied. For this purpose, adhesive dots were applied on anode sheets of a pouch cell

manufactured according to chapter 4. This method has been proven by Cannarella et al. and our earlier publication to lower ion transport at these glued locations (visualized in Supplementary Fig. 1)[25,26]. According to Cannarella et al., this results in an increased ion flux at the edges or weak spots of the adhesive dots, leading to overpotentials and lithium plating. The adhesive pattern throughout the electrode stack can be viewed in Supplementary Fig. 5. To obtain the lowest possible ultrasound attenuation and, therefore, the highest signal amplitude, this effect will be visualized on the first anode layer. This leads to the sensors maintaining a distance of exactly 1.885 cm to the cell surface. The distance is based on measurements of cell housing, separators, and electrode thicknesses. While deeper layers can be visualized, image quality may be inferior when using 25 MHz transducers compared to lower frequency transducers (see Supplementary Fig. 6 for a comparison of visualization at the first and last layers of a 4 mm thick cell). Figure 1b shows a representative reflected ultrasound wave. It is evident that the signal amplitude, denoted as $A_r$, is initially high, and even clipping, until about 0.2 µs and decreases over time due to sound attenuation within the cell being a function of distance. Since the first signal part only represents the reflection at the cell housing, it is not further analyzed in this work. Using the cell's speed of sound and the sensor bandwidth, the region in the signal representing an interaction at the focal point can be calculated. According to the data sheet of the sensor, this point starts at -0.38 µs. Since the bandwidth of the considered signal in the time domain is $t_b = 0.21$ µs, the range from 0.38 µs to 0.59 µs is examined. It is common to divide the signal into segments (so-called gates) of the length of the bandwidth in the time domain and analyze these segments separately[27]. This approach allows achieving layer resolution with sufficient spatial resolution of adjacent layers near the focus point[22]. By integrating the ultrasound signal within a gate, a scalar value is formed, which can be assigned to a colormap (see Fig. 1c). Bright values indicate a strong reflection at the focal point, while dark values indicate a low reflection amplitude. According to Wasylowski et al.[22] and Bauermann et al.[27], the strength of the reflection depends on the difference in the acoustic impedances of adjacent materials at the interface. Repeating this process at each point (resulting in a pixel) of a cell or multiple cells, as indicated in Fig. 1d, leads to the generation of an image as shown in Fig. 1e.

### Ultrasound analysis
To visualize lithium plating in real-time, we used three separate cells, each manufactured according to the protocol described in section "Methods". The goal is to visualize the lithium plating provoked by the adhesive dots on the anode surface during cycling. The following analysis is focused on the individual scanning of cell 1 and cell 2. A third cell (called cell 0) was used for preliminary examination. For cell 0, we scanned all cells, including the cell holder, while only cycling cell 0. The results of this preliminary experiment are shown in Supplementary Fig. 2. The results of the primary measurement series for cell 1 are shown in Fig. 2a. In the first series of measurements, a CC-CV charge was carried out at a charge rate of 1C and a CV phase up to C/10 charge rate (see Fig. 2b). Further information regarding the electrochemical procedure and equipment can be found in section "Methods". During cycling, an ultrasound image is recorded every 56 s. The sequencing of images into a video can be seen in Supplementary Movie 1 and Movie 2 (for both cell 1 and cell 2). For the presentation of the results in the manuscript, six key points in time in the charging process are shown. In state (1), the cell is fully discharged at 3 V without prior cycling (after formation, degassing, shipping, and discharging to 3 V) in the pristine state. In state (2), the cell voltage has increased to 3.76 V. The brightness of the ultrasonic image has become slightly higher, which can be explained by the phase transitions and expansion of the anode during the charging process[28]. These change the acoustic properties, such as attenuation and sound velocity of the ultrasonic wave through the cell, and thus lead to a slightly stronger reflection in the focal point[29]. Based

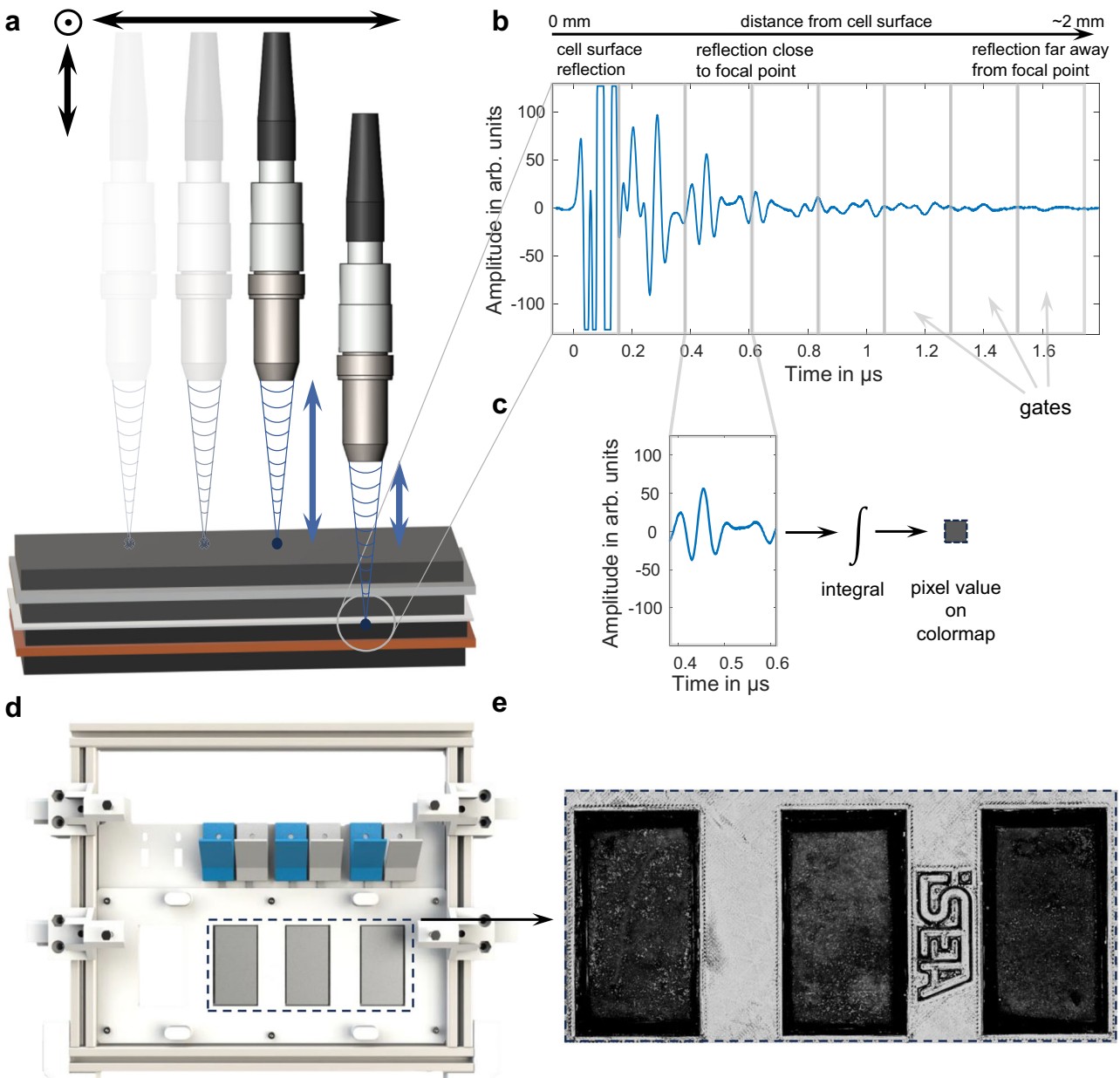

**Fig. 1 | Working principle and signal processing of the ultrasonic imaging method. a** Displays the discrete movement necessary to scan the cell. One ultrasound measurement is taken at each point. Information from deeper layers can be extracted by moving the sensor closer to the cell, thereby moving the focal point. **b** Showcases an exemplary ultrasound signal segmented into equal time gates. Each subsequent gate captures signal portions originating from deeper within the cell (up to 2 mm without moving the sensor, depending on the speed of sound and attenuation), while the initial gates display reflections from the cell's surface or its

proximity. The gates are designed based on the signal bandwidth, which results in gate 3 capturing the reflection at the focal point. **c** Displays the method to visualize one pixel of the raw data. All gates are individually processed in real time by integrating the absolute value of the signal. The result is then mapped onto a colormap for visual representation of the pixel. This process is repeated for each spot of the cell holder for up to four cells (three loaded in example **d**), resulting in the ultrasound image (**e**).

on the findings of Hsieh et al. and Davies et al., it can be assumed that changes in attenuation have a highly similar effect on the transmission and reflection amplitude during cycling[30,31]. A detailed explanation concerning the increase in reflection amplitude between states (1) and (2) can be found in the Supplementary Material. In state (3), the cell voltage has reached 4 V. Here, a bright white spot is visible for the first time in the lower right corner next to the cathode tab. As described in chapter 2.1, this can only be caused by a sudden change in the acoustic impedance of the electrode or by adding an additional deposition layer with significantly different acoustic properties. The transmitted amplitude is assumed to be decreased because of the increased

reflection caused by the Li deposits[29] In state (4), the cell has reached 4.16 V. Two more bright white spots have already appeared here. The spot near the cathode tab has become even clearer. In state (5), the cell has reached the end-of-charge voltage, and the spots have reached their maximum size. In state (6), the cell is fully charged and starts relaxing for 10 min. The spots have become slightly smaller in this state. 10 min after charging, the cell was extracted from the setup for an ex-situ examination (see chapter 2.3). Based on the dynamic behavior of the white indications in the ultrasound images, it can be suggested that they seem to represent characteristics similar to lithium plating.

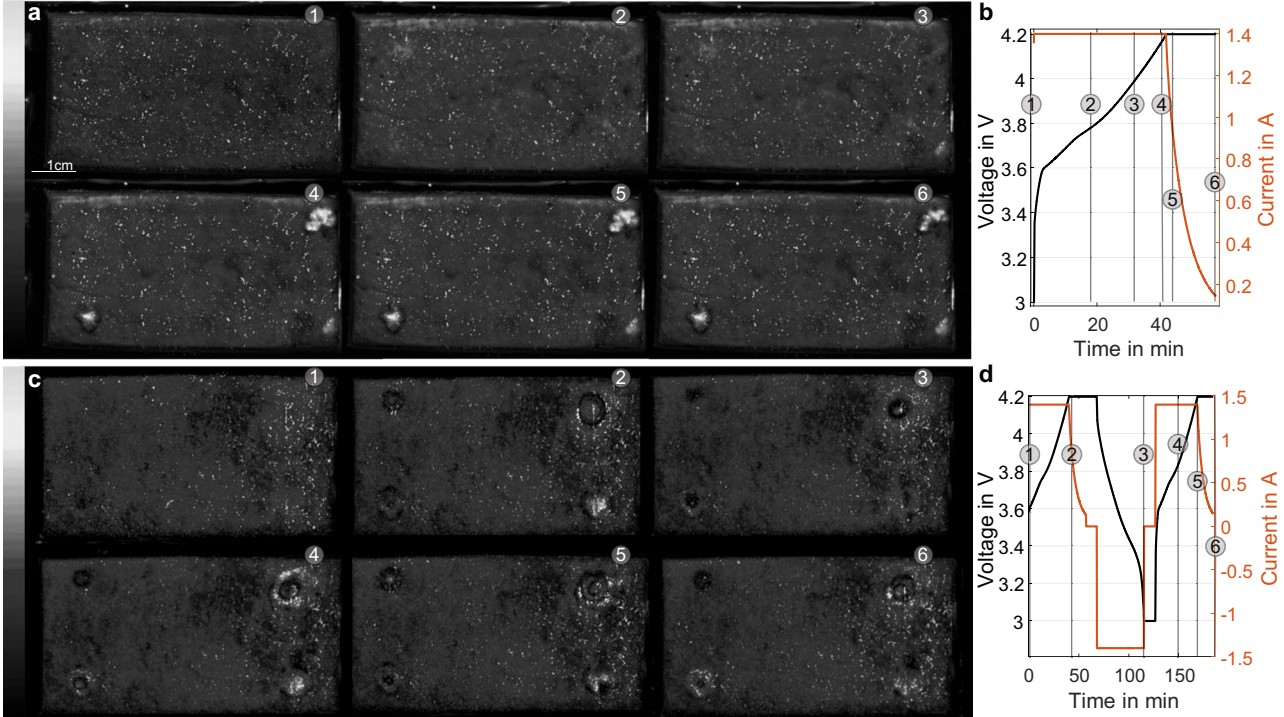

**Fig. 2 | Selection of operando ultrasound images and electrochemical data at significant points during cycling of two pouch cells.** The tabs are located on the right side of the cell (not displayed). **a, b** Portray the charging procedure of cell 1. At points (1) and (2), the ultrasound images are relatively homogenous. A circular indication starts to appear at point (3) in the lower right corner. At points (4)–(6), more indications appear, which first get larger and then decrease in size shortly after the CV phase begins. **c, d** Display the first cycles of cell 2. While similar indications appear on the ultrasound images during charging (1)–(2), they decrease even more during discharging (3) and increase again during the next charging phase (4)–(5). After the CV phase, the indications decrease again (6). Additionally, the plating seems to appear mostly on the edges of the adhesive dots of cell 2. We refer to Supplementary Movies 1 and 2 for the videos based on all ultrasound images. Source data are provided as a Source Data file.

In the second series of measurements, the aim is to investigate whether the appearance and contraction of the bright indications in the ultrasound images are reproducible from cycle to cycle and for a different cell. For this purpose, another discharging and charging step was added to the measurement protocol with a second cell (cell 2). Once again, six distinctive points in the measurement process are shown (see Fig. 2c, d). At the starting point (1), the cell is again in its pristine state. In state (2), the CV phase of the first charging process has already begun. Distinctive indications are visible here, similar to cell 1, but some appear around the adhesive dots instead of on top of them. At point (3), the cell is again discharged to its cut-off voltage. Similar to state (6) in Fig. 2a, the spots not only become smaller but also darker. According to Hsieh et al., this indicates that the active material at this point again assumes similar properties to the rest of the electrode[30]. In particular, the spot near the cathode tab (bottom right) has become much less visible. In state (4), the cell was charged again to a cell voltage of 3.82 V. In this state, new bright spots are visible, some of which form rings around the previously existing indications. It should be noted that these spots appear earlier than during the first charge and lead to an overall larger indication formation. During and at the end of the CV phase in states (5) and (6), the indications become smaller again but reach a larger size than in the first CV charge (comparison (2)–(5)).

Based on the dynamics of the indications occurring in the ultrasound images, we hypothesize that these are provoked by lithium plating. This hypothesis is initially motivated by the repeatedly confirmed spot-like plating formation in the literature[32]. In addition, the shrinking of the indications in the ultrasound image during relaxation and discharge is consistent in speed and extent with current results in the literature concerning lithium plating dynamics[33]. In addition, the fact that the dynamics and the extent of the bright indications in the ultrasound image are more pronounced near the electrode tabs (on the right side of the images) is a further sign of lithium plating activity[34].

Two methods are used to verify our hypothesis in chapter 2.3:

1. Reference electrode (RE) measurements: This is used to verify the cell voltage during the charging processes at which the anode potential stays below 0 V vs Li/Li$^+$ and lithium plating is thermodynamically feasible[35].
2. Comprehensive post-mortem analysis: This is to verify plating in terms of location, size, shape, and morphology.

**Verification**

In order to verify whether the spot formation and contraction are actually caused by lithium plating, a separate set of cells with an RE was manufactured. The description of the production of these can be found in chapter 4. A rendering of the cell with an RE can be found in Fig. 3a. Figure 3b shows the voltage and anode potential data during the cycling of a representative RE cell. The results are representative of a set of six RE cells, all presenting comparable data. Here, the cell was cycled starting at 3.68 V with a current rate of 1C according to the CC-CV protocol from chapter 4. It can be seen that the anode potential measured with the RE falls below 0 V at the first charge at a cell voltage of 4.02 V (taking into account the measurement accuracy of the voltage measurement from chapter 4). In order to quantify the potential shift due to polarization, and in order to make the cycling protocol identical to Fig. 2d, a 10 min break was inserted after each charge and discharge process. Due to the relaxation of the anode potential, the anode voltage drops below 0 V at a cell voltage of 3.97 V, taking into

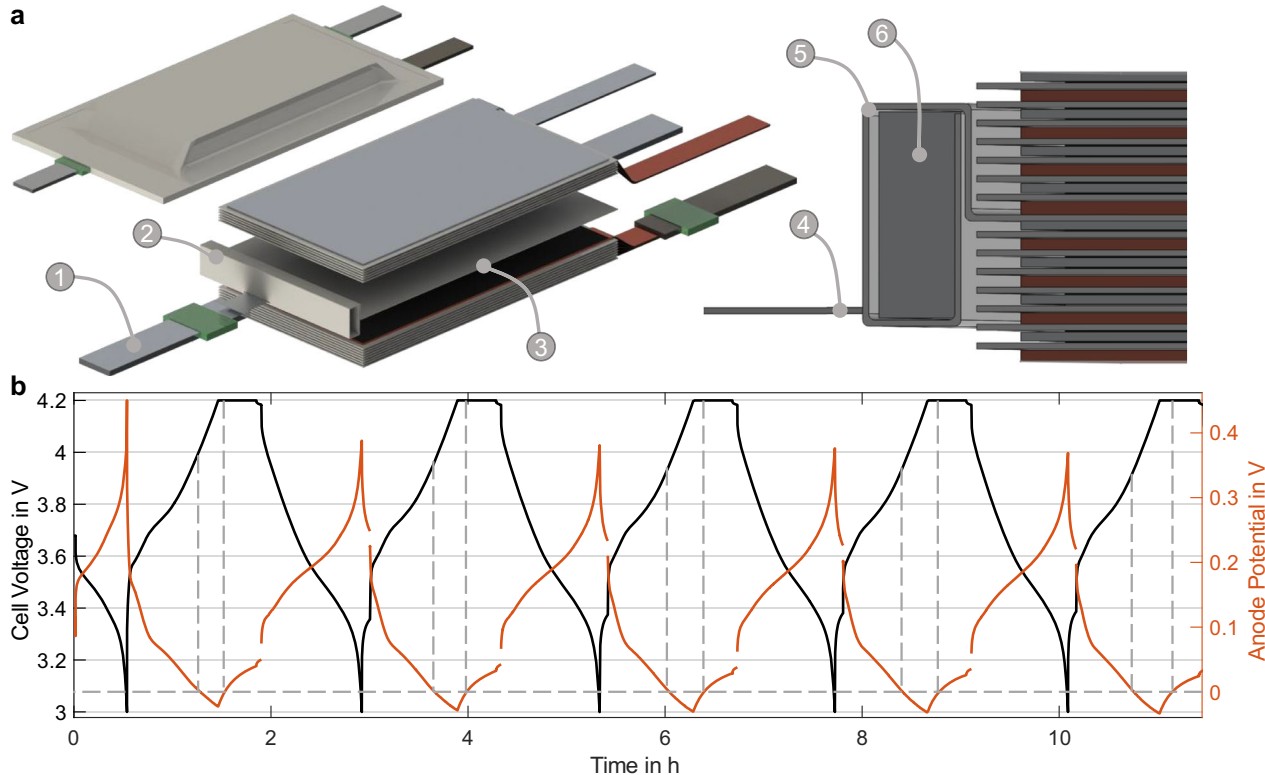

**Fig. 3 | Verification results of reference electrode (RE) measurements.**
**a** Renderings of cell with reference electrode and placement. For better visibility, only 7 out of 15 layers are displayed. (1) RE tab with sealing for connection of voltage measurement equipment. (2) RE made from LTO wrapped in separator material of an elongated separator from the electrode stack. (3) Part of an elongated separator that is placed inside the electrode stack instead of the regular-sized separator. (4) Nickel-plated copper connector welded to the RE. (5) Separator winded around RE. (6) LTO-based RE (side view). **b** Cell voltage and anode potential during 1C CC-CV cycling. Voltages where the anode potential drops below 0 V are marked with a dashed line. These results are in line with the findings from Fig. 2. Source data are provided as a Source Data file.

account the measurement accuracies. Furthermore, the voltage at which the anode potential drops below 0 V slightly decreases from cycle to cycle. Overall, the ranges in the cell voltage at which the anode potential drops below 0 V match with the results from Fig. 2, which suggests that the ultrasound indications are, in fact, lithium plating. However, measuring the anode potential with reference electrodes is subject to errors and inaccuracies. For example, local differences in the anode potential are not measured, but, as already shown experimentally and simulatively by Oehler et al. for the same RE used in this work, only a so-called mixed potential is measured. This potential is strongly influenced by the potential at the edge of the electrode and the anode overhang[36]. Furthermore, due to the electrolyte resistance, polarization, and tortuosity of the separator, it is not trivial to quantitatively detect the occurrence of lithium plating based on the measurement of RE alone. However, since the locations where lithium plating was provoked are relatively close to the anode overhang, it can be assumed that uncertainty is in the range of about 15 mV (overestimation), as measured in ref. 36 with a similar configuration. Nevertheless, the temporal correlation of the RE measurement with the ultrasound images is remarkable: The indications in the ultrasound images start to appear very close to the same cell voltage where the anode potential drops below 0 V and begin to shrink when the anode potential is above 0 V.

In order to obtain direct ex-situ evidence of lithium plating in addition to the temporal correlation, a comprehensive post-mortem analysis was carried out immediately after the charging of cell 1 from Fig. 2 (at 4.2 V after the CV phase). For this purpose, a cell opening and disassembly was performed as a first step. The processes that were carried out for this are described in chapter 4. The anode sheet at

which the ultrasound wave was focused is shown in Fig. 4a (front side) and b (backside, flipped over at the long edge). Direct locational, geometric, and dimensional correlations between the markings (1) and (3)–(6) and Fig. 2a are clearly recognizable. There are, among others, residues of the adhesive visible, which was used to provoke local plating (see chapter 2.1 and 4). Markers (1), (5), and (6) also show silver deposits on the circular adhesive dots. Marking (3), on the other hand, is matt and not very noticeable. Furthermore, in marking (4) of Fig. 4a, b (8) silver deposits are visible. Marking (7) shows an inconspicuous area of the anode material as a reference. In addition, a gray deposition is visible in marking (2) as an example. This gray deposit is congruent with the Y-shaped one on the back side in Fig. 4b marked with (9). At this point, we refer to Supplementary Fig. 3a, b, where gate 4 (unlike gate 3 in Fig. 2) of cell 1 has been visualized and in which the gray structures are already visible before charging begins. Supplementary Fig. 3c, d also show that the gray depositions on the anode surface are not present on cell 2 and are also not visible in the ultrasound images of cell 2 when analyzing gate 4. In Fig. 4c, d, we present a direct comparison of the optical and ultrasound images of cells 1 and 2. To achieve this, we superimposed the optical images of both sides of the first layer anode in Fig. 4c, along with the resulting ultrasound images based on the analysis of both gates 3 and 4 in Fig. 4d. Despite the optical images being taken post-mortem, this correlation between both imaging methods remains noteworthy.

To confirm the presence of Li plating around or above the circular adhesive dots and to differentiate the grayish Y-shaped structures in the center of the cell (which were already observed via ultrasound before charging) from Li plating, we extracted samples from markings (2), (6), (8), and (9). These were then examined using SEM-EDX (see

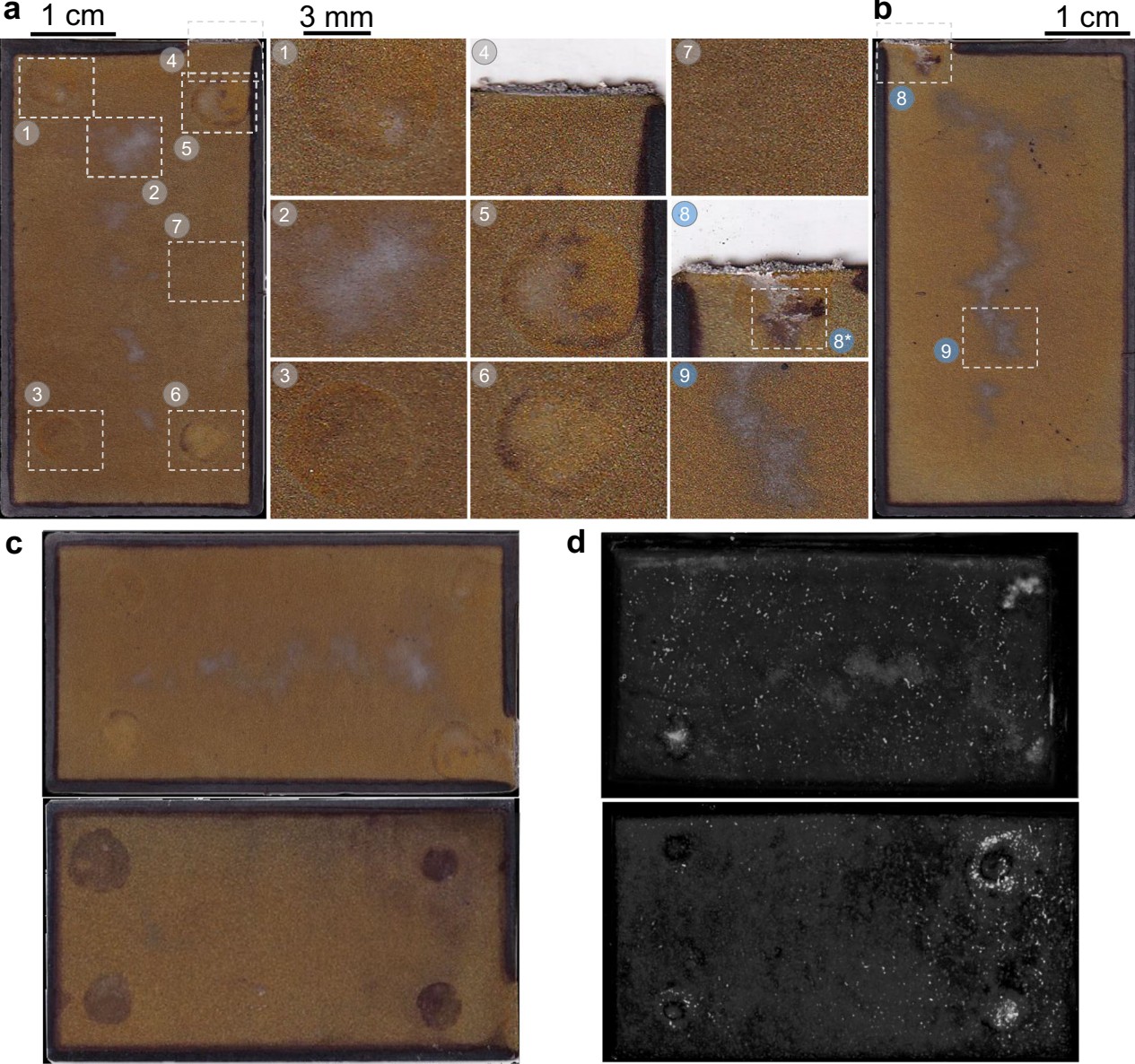

**Fig. 4 | Optical images of the first anode layer visualized by ultrasound.** The images were extracted during the post-mortem analysis of cell 1 at a cell voltage of 4.2 V. **a** Shows the front side of the anode sheet. Gray markings represent samples from the front side of the sheet, while blue markings indicate samples from the back. Markings (1), (3)–(6) indicate silver depositions near or on top of the adhesive dots. Marking (2) tags a gray deposition in the center of the anode sheet. This sheet is partly congruent with the Y-shaped deposition on the backside of the same anode sheet in (**b**) marked with (9). Marking (7) tags a reference area of the anode.

Markings (8) and (8*) show a large agglomeration of silver depositions near the cathode tab as well as an adhesive dot next to it. The silver depositions are in line with the bright indications in the ultrasound images. **c**, **d** illustrate optical and ultrasound images of cell 1 and 2 for direct comparison. For the optical images in (**c**), we transparently overlayed images from the front and back side of the first anode layer. For the ultrasound images in (**d**), we transparently overlayed the images generated by analyzing gate 3 and gate 4.

chapter 4 for details regarding the devices used). The resulting SEM images and EDX spectra are shown in Fig. 5. In the first column, the results of the sample (6) are shown with different zooms on the silver depositions. The resulting EDX spectrum reveals an atomic metallic lithium content of 85% in the blue marked area in the second row. A zoom in the first row shows that the metallic lithium components represent the accumulations of metallic lithium on the graphite flakes. These correspond to the morphology type B of the publication by Kühnle et al.[37]. The second column in Fig. 5 shows SEM and EDX results of the grayish deposits in the middle of the cells. In contrast to sample (6), there are no circular lithium metal deposits on the graphite flakes but patchy deposits, which, according to the EDX spectrum, exhibit a high amount of fluorine, a smaller amount of phosphorus, and no

lithium. Various publications show that deposits of this morphology and composition could represent an SEI formed during the formation cycles[38,39]. However, the phosphorus content also indicates that these may be reaction products with the $LiPF_6$ conducting salt used or the PVDF binder, especially because these structures were already visible in the ultrasound image before the first charge after forming (see Supplementary Fig. 3). To keep the focus of this work on lithium plating detection, further explanation regarding the formation of the gray deposits in the middle of cell 1 is provided in the Supplementary Material. The third column shows a zoom on marking (8) and (8*), both of which have silver deposits on the edge of the anode (8) and on an adhesive dot (8*). Once again, the circular accumulations can be found (especially in the zoom in the first row), which have an atomic metallic

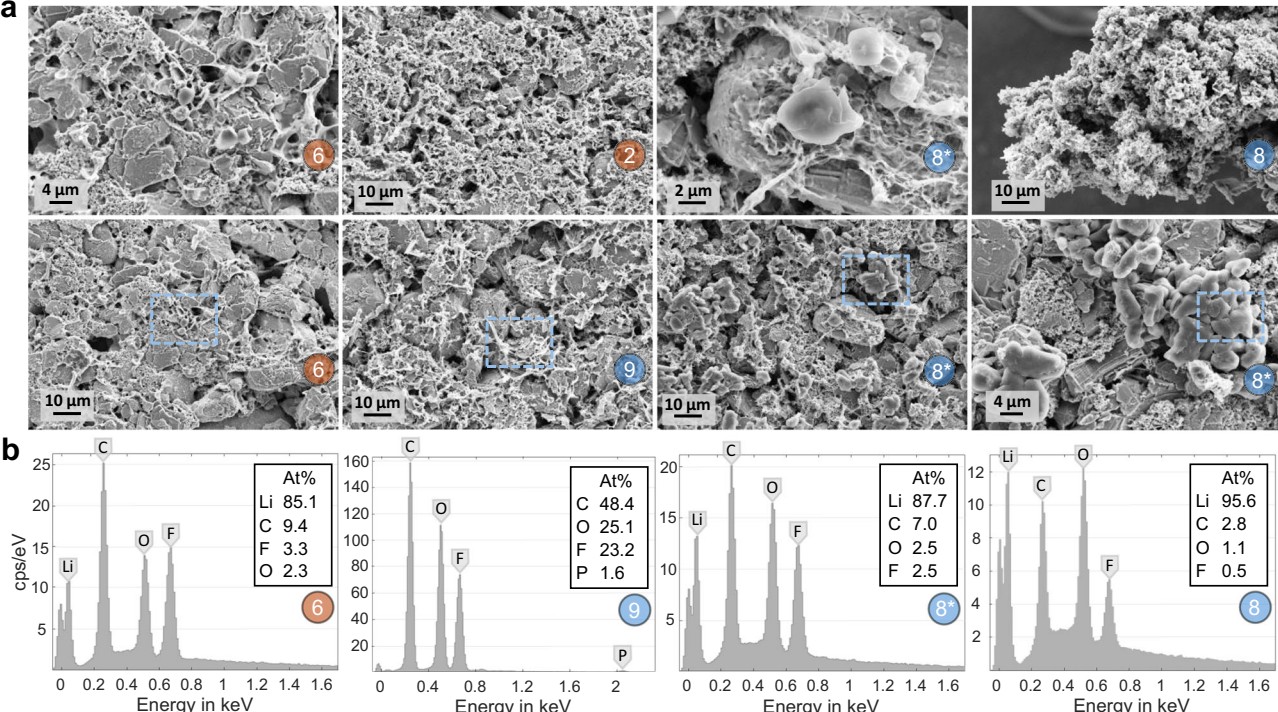

**Fig. 5 | SEM-EDX results for lithium plating verification. a** shows SEM images of the samples from markings (2), (6), (8), and (9). Red markings represent samples from the front side of the sheet, while blue markings indicate samples from the back. **b** shows the corresponding EDX results from representative positions of the samples (At% refers to the atomic fraction of each element within the marked sample). Based on the morphology in the SEM images and EDX analysis, it can be concluded that metallic lithium is present at the adhesive spots. The gray deposits located in the center of the electrode are featured in the second column. While no evidence of metallic lithium plating was detected in this area, it is highly probable that a reaction product of the electrolyte is present. This aligns with the findings observed in the ultrasound images.

lithium content of 87.7%. The last column shows results from the sample (8). In the first row, there is an SEM image of the silver deposit hanging over the upper edge of the anode. The morphology of this deposit corresponds to type A of the publication by Kühnle et al.[37], and thus strongly indicates lithium plating. This is confirmed by the EDX spectrum in the last row, which shows an atomic metallic lithium content of 95.6%. In the second row, a larger accumulation of spherical lithium plating from the sample (8*) is shown, which indicates that there is a particularly strong occurrence of lithium plating at this point, which is consistent with the optical images.

Based on the SEM images and the EDX analysis, it can be stated that lithium plating has formed on top and around the adhesive dots on cell 1. This is in line with previous publications from refs. 26,40. According to Canarella et al., it can be assumed that the adhesive dots have restricted lithium transport in cell 1 and led to an overpotential, which confirms our hypothesis of the occurrence of lithium plating. Based on this analysis and in combination with the optical images from Supplementary Fig. 4b, it is apparent that lithium-ion transport was completely blocked by the adhesive in cell 2 in contrast to only being restricted in cell 1. This results in the circular plating behavior around the adhesive dots visible in Fig. 2c and no lithiation of the graphite under the adhesive dots of cell 2. Furthermore, this confirms the earlier occurrence of lithium plating in cell 2 compared to cell 1 (see Supplementary Movie 2), as a more inhomogeneous ion flux density due to the completely blocking adhesive leads to an increased overpotential.

Overall, the SEM-EDX analysis is consistent with the optical and ultrasound images, as well as the reference electrode measurements. An increased ultrasound reflection, which would result in a brighter pixel when analyzing gate 1 instead of gate 3, could, in principle, also be caused by local gas generation. However, a comparison of ultrasound images with the same sensor selection and the same measurement settings shows a different ultrasound image at gassed locations of other comparable cells (see Supplementary Fig. 7). Due to the partially reversible size and shape of the indications in correlation with the electrochemical data, it is also not assumed that the indications represent another failure mode such as electrolyte dry out[41]. Based on the extensive verification, a convincing database has been created on which, in combination with information from the literature, it can be safely stated that the spots in the ultrasound image visualize the formation and stripping of lithium plating. This is evidenced by the spatial, temporal, and morphological correspondence of the indications. Based on the similar dynamics in the ultrasound images, findings from this comprehensive analysis can be safely transferred to cells 0, 2 as well as other cells in pouch or prismatic format. We, for the first time in literature, illustrate that even subtle variations in cell behavior, leading to modified plating dynamics, can be distinctly visualized without resorting to neutron-based techniques. Moreover, this ultrasound visualization reaches a spatial resolution of 75 μm in real-time, which was previously unattained in the literature.

## Discussion
In this work, we present a procedure for real-time visualization of the internal dynamics of battery cells using ultrasonic imaging, demonstrating a significant advancement of this operando method. The images were sequenced into a video, with each pixel spanning a width of 75 μm, being the highest-resolution ultrasound imaging published. The visualization was carried out without the use of neutron-based methods and utilized low-cost hardware, enabling widespread use and scalability. We demonstrated the functionality of our approach using a set of three pouch cells, which were cycled under 1C CC-CV conditions. We successfully identified depositions in real-time within multiple cells, conclusively confirmed as lithium plating through an extensive

verification process. The imaging method was automatically tuned so that other deposits that were not lithium plating could be deliberately separated by shifting the focal point of the ultrasound sensor to the point of interest. Separately manufactured cells with reference electrodes have verified the temporal occurrence, growth, and stripping of the ultrasonically visualized lithium plating by the under- or overshooting of the anode potential of 0 V vs Li/Li⁺. Furthermore, we proved the occurrence of metallic lithium at the areas identified with ultrasound imaging in an SEM-EDX analysis using a detector capable of measuring ultra-soft Li $K\alpha$ X-rays at around 54 eV. This technique enables design flaws in battery cell manufacturing to be detected early and cost-effectively without disassembling cells or other complex procedures. Furthermore, this method facilitates the design of fast charging processes with 2D information in such a way that the cell does not exhibit metal plating using the indications in the ultrasound image as a feedback loop. Since the focal point of the ultrasound wave can be set automatically, this method does not require parameterization like physico-chemical approaches such as the Newman model. This significantly reduces the barrier to entry for spatially resolved operando aging studies and enables parallelization in cell development due to the low cost of the ultrasound imaging hardware requirements. While current commercial ultrasound devices are offered at a high price, the focus of this work is on demonstrating the potential of the measurement method. Work on the low-cost implementation of the measurement method can thus help to promote widespread implementation of the technology. Current limitations of this method include the visualization of effects within the cell during cell clamping, which was not considered in this publication. Clamping battery cells adds interfaces between plates and cell surfaces, leading to artifacts (Eqs. (1) and (2)). This can be compensated by using suitable clamping material like Rexolite.[42] Another limitation is the use of a coupling agent (e.g., distilled water or silicone oil), which induces immersion cooling, potentially deviating from real-world conditions. This can be circumvented by employing water-based compression pads (e.g., as proposed in ref. 43). Alternatively, based on the introduction of a water jet in front of the sensor and direct suction at the cell, it can be ensured that the specimen only comes into contact with liquids at the sensor location. If the use of water is to be avoided, alternative liquids such as mineral or silicone oil can be used. Alternatively, laser-induced ultrasound can be used to avoid the use of liquid altogether[44]. Since we have actively tried to circumvent other aging effects in this experiment, a further open research question is how the presented method could be used to quantitatively distinguish between overlapping aging mechanisms. For example, a combination of transmission and reflection measurements would be feasible to exploit different sensitivities of the modes to other aging effects. Furthermore, the compensation of SoC- or temperature-induced amplitude changes through cycling at high current rates should be considered in future work, as this can result in amplitude errors of up to 3%[45,46]. Another potential area of application of the presented method using higher resolution ultrasonic sensors would be the real-time investigation of macroscopic plating morphologies as proposed by Kühnle et al.[37]. These limitations will be addressed in future work.

In conclusion, the data and analysis of this work provide a comprehensive tool to enable the transfer and use of operando ultrasound imaging in various areas of cell development and manufacturing.

## Methods
### Cell manufacturing and materials
The battery cells presented in this work contain commercial NMC622 HED (manufactured by BASF SE, Germany) as cathode material and commercial graphite (manufactured by Showa Denko K.K., Japan) as anode material. The cathode has a capacity density of 3.0 mAh/cm², the anode of 3.4 mAh/cm². The dimensions of the cathode are 29 × 55 mm (WxL) and of the anode 31 × 56 mm (WxL). The separator

measures 65 × 59 mm (WxL). It has been folded once along the longer side to create a pocket. The anode was subsequently placed inside the separator pocket (see Supplementary Fig. 1d) and fixated using a PVDF and acetone-based adhesive. As explained in chapter 2.1 this is intended to provoke lithium plating due to reduced local conductivity. The cells are stacked by hand in the pilot line at Fraunhofer ISIT, Germany, and consist of 15 double-coated galvanic units (or layers). The anode surface is 1mm larger on each side than the cathode. A commercial separator Celgard 2325 (manufactured by Celgard LLC, USA) and a commercial pouch foil DNP-ALF (manufactured by Dai Nippon Printing Co. Ltd., Japan) were used. The cell stack is about 7mm thick. The cells were filled with 8 ml commercial LP57 + 2% vinyl carbonate electrolyte (manufactured by UBE-Industries Ltd., Japan). All cells were formatted with two C/10 (0.14 A) cycles, one C/5 (0.28 A), and one 1C (1.4 A) cycle in the range of 3.0 V to 4.2 V. Renderings of the cell design can be viewed in Supplementary Fig. 1. More information on the electrodes and cell performance can be found in Supplementary Table 1.

### Electrochemical cycling
Due to the relatively low capacity of the battery cells of 1.4 Ah, the potentiostat NGU 202 (manufactured by Rhode&Schwarz GmbH & Co. KG, Germany) was used to cycle the battery cells. It has a voltage accuracy of <0.02% + 1 mV (accuracy + offset) and a current accuracy of at least (<0.025% + 500 µA) for the actuating variables. With regard to the measured variables, it has a voltage accuracy of <0.02% + 500 µV and a current accuracy of <0.025% + 250 µA. All cells were connected with a four-point contact via the sense line of the potentiostat for voltage measurements. The cells were cycled with 1C (1.4 A) in the voltage range of 3.0 V to 4.2 V specified by Fraunhofer ISIT. A break-off current of C/10 (0.14 A) was specified for the cut-off criterion of the CV phase. After each charge or discharge, a 10-min pause was applied, during which the cell was disconnected from the potentiostat via relays. All cycling experiments were performed at a temperature of 20 °C inside of the ultrasound imaging apparatus. The temperature was controlled using the water heating/cooling function included in the Sonoscan (D9650 C-SAM).

### Ultrasound generation and signal analysis
For the generation of the ultrasound signal and the movement of the sensors, a commercial system from Sonoscan (D9650 C-SAM) was used. Additionally, an ultrasound sensor from the company Sonoscan was used. It has a resonance frequency of 25 MHz, a focal length of 1.905 cm, and a bandwidth of 100%. The diameter of the piezoceramic is 6 mm. The sensor is excited with a rectangular pulse of ~20 ns width and 400 V to generate a sound wave. This wave with amplitude ($A_i$) then travels through distilled water (coupling medium) to the battery cell, where it is partially reflected and transmitted. The conductivity of the distilled water was constantly measured to be ≤1.2 µS/cm. The reflected ($A_r$) wave amplitudes at each material interface are described by the following equations.

$$A_r = \left(\frac{Z_2 - Z_1}{Z_1 + Z_2}\right)A_i \tag{1}$$

with

$$Z_i = \rho_i\left(K_i + \frac{4}{3}G_i\right) \text{ with } i = 1, 2. \tag{2}$$

$K_i$ and $G_i$ are the compression and shear moduli, respectively, which can be calculated from the modulus of elasticity $E_i$ using Poisson's ratio, and $\rho_i$ the density (for $i = 1, 2$). The focus point is positioned so that it hits the correct layer based on the layer thickness specified in Supplementary Table 1 and the pouch foil and separator thickness specified by the manufacturer. To keep the focal point on an electrode

sheet independent of surface effects, a constant time-of-flight (ToF) of 27.16 μs was set. The Sonoscan gantry then maintains this ToF during the scan to keep a constant distance to the cell surface. In this work, only the reflected signal is measured. In the reflected signal, the portion in the time domain that lies in the calculated focal point based on the speed of sound and bandwidth is then analyzed. The visualization is done by integrating the absolute value of the signal in the previously calculated time range and mapping it to a colormap (grayscale in this case). When the focused ultrasound beam encounters a point covered by a material with a significantly higher acoustic impedance than the graphite electrode, such as metallic lithium, it forces an amplified reflection as per Eq. (1). This amplification consequently enhances the pixel brightness in the resulting ultrasound image according to chapter 2.1.

### Reference electrode measurements

All cells in this work were build according to chapter 4.1. To measure the anode potential for verification in chapter 2.3, a separate set of cells was additionally manufactured with an LTO reference electrode. The LTO RE was first wrapped into a separator (manufactured by Whatman plc, UK) and then was attached to the bottom edge of the cell (visible in Fig. 3). For this purpose, the reference electrode was additionally wrapped with the middle separator of the cell stack, which is 15 mm longer than the other separators, and then fixed carefully at the bottom of the cell using Kapton tape. Positioning the RE outside the cell stack minimizes the variation of the current density in the cell and thus leads to a comparable performance to a cell without a reference. The comparability of the electrochemical performance in such a configuration has also been confirmed in other work[47]. According to ref. 48, this method does not lead to any changes in the plating behavior. A nickel-plated copper contact was welded to the electrode, to which the tab is welded (see Fig. 3).

### Post mortem analysis

The post-mortem analysis was performed 10 min after the last ultrasound image. Cell opening and sample preparation were done in an argon-filled glove box with low $O_2$ (<3 ppm) and $H_2O$ (<0.3 ppm) values, to ensure that the samples do not react with the environment. Immediately after opening and separation of the cell components, the electrodes (anode/cathode) were analyzed with a flatbed scanner (Canon CanoScan LiDE 300) inside the glovebox. The scan resolution was set to 600 DPI and thus a pixel size of 42 μm. The scanning electron microscope (SEM) analysis was performed with a Zeiss Supra 55 using the SE detector and an accelerating voltage of 5 kV. The SEM samples were prepared directly after the cell opening in the argon-filled glove box and then transferred to the electron microscope using a transfer module (Kammrath & Weiss GmbH). With the transfer shuttle, the specimens were protected from oxygen and moisture at all times, mitigating the risk of unwanted reactions during the transfer. For the energy-dispersive X-ray spectroscopy (EDX) measurements, the Oxford Instruments Ultim Extreme was used. With the detector, it is possible to detect the very weak lithium signal (ultra-soft Li Kα X-rays at around 54 eV) directly in the spectrum. Therefore, it is possible to validate the elemental compositions of the lithium-plated surface structures.

## Data availability

The electrochemical and ultrasound image data generated in this study are provided in the Supplementary Information/Source Data file. Source data are provided with this paper.

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

## Acknowledgements
This work was funded by the Federal Ministry of Education and Research as part of the research cluster "Batterienutzungskonzepte" through the project MADAM4Life (grant number 03XP0327C, D.U.S.) and NUBase (grant number 03XP0322C, D.U.S.) and as part of the research cluster "AQua" through the project InOPlaBat (grant number 03XP0352A, D.U.S.). Open Access funding enabled and organized by Projekt DEAL.

## Author contributions
D.W. performed the experimental studies, carried out the analysis, and served as the project lead. H.D. and D.W. jointly carried out the cell opening. H.D. performed the SEM-EDX analysis. M.S. contributed to the analysis of the post-mortem data. L.L. and T.F. supported the visualization of the findings. E.R. performed the cell manufacturing. A.B., A.W., F.R., and D.U.S. contributed to the funding acquisition as well as proofreading of the manuscript.

## Funding

## Competing interests
The authors declare no competing interests.
