## [Transparent Peer Review file · Nature Communications]

Operando Visualisation of Lithium Plating by Ultrasound Imaging of Battery Cells

Corresponding Author: Mr David Wasylowski

Version 0:

Reviewer comments:

Reviewer #1

(Remarks to the Author)

This review is for the submitted paper "Operando Visualization of Lithium Plating by Ultrasound Imaging of Battery Cells" by Wasylowski et al. While development of ultrasound for non-destructive characterization of batteries has been well substantiated in prior studies, I believe the current work is of potential significant novelty for the field and I recommend publication in Nature Communications pending the revisions as described below. One of the key critiques of ultrasound, in comparison to neutron imaging or X-rays, in the detection of Li plating for fast charging is the lack of spatial resolution, and in particular, the inability to resolve degradation within an individual electrode layer for a multilayered closed form battery. While one might argue that exact localization of Li plating is not necessarily needed, since gauging SOC/SOH may be sufficient in determining a bad cell, pushing the boundaries of what ultrasound can achieve is still an important research need. This paper begins to show that local lithium deposition, not just spatially across a battery, but also within the depth of the cell stack, can be achieved with sufficient signal processing and advanced scanning acoustic microscopy hardware.

- p. 5: the authors state that the brightness becomes higher during charge. My understanding is that this brightness is the intensity of the amplitude that is reflected back. However, previous studies using transmission mode indicate an increase in transmission amplitude upon upon charging. Should the pulse echo changes be opposite that of the transmission mode? (waves that transmit better should have poorer echo intensity).
- on the other hand, the authors' explanation that the bright white spots (p. 6) indicate Li plating appears to make sense, as thick Li deposits are likely to induce transmission amplitude attenuation, and thus greater wave reflections. is this correct?
- were the adhesives placed on ALL anode sheets, or just the anode sheet closest to the transducer?
- Fig 4 the authors depict the optical image of the anode sheet that is closest to the focusing length of the transducer. What is this length and which anode sheet is this? I think it would be more clear if the authors put a few acoustic images next to the same optical images for direct comparison.
- Further, for a complete demonstration, the authors should then change the focus length and show another comparison of another electrode layer further away from the transmitting transducer. is this possible, or is it only possible to look at the first few layers? This part is particularly important to address. I imagine a 25 MHz wave cannot penetrate too deeply. What does it mean if the technique can only see the first few layers - do we need lower frequency transducers? Along the same lines, these frequencies are still longer wavelengths than individual electrode layers (and therefore, waves could overlap between layers and/or exhibit complicated partial wave reflections).
- how do the authors guarantee there is absolutely no gas formation or electrolyte dry out in their cells? how does one decouple these failure modes over longer cycles? e.g. [https://www.cell.com/joule/pdf/S2542-4351\(23\)00440-3.pdf](https://www.cell.com/joule/pdf/S2542-4351(23)00440-3.pdf) demonstrated that spatial heterogeneities were due to electrolyte consumption
- For the SEM/EDS, were the samples thoroughly rinsed in solvent and dried before imaging? Regarding the statement about reaction with LiPF6 - I'm not sure if this is a common belief in battery literature, that the dissolved LiPF6 reacts (without any applied current/potential). Solvent reduction should be the major degradation mode.
- In the conclusion the authors state the hardware is low cost as a benefit. The Sonoscan is at least \$200K USD - is this still "low cost" or "enabling widespread use and scalability"? Would any battery manufacturers be willing to submerge their large cells in water (even though its low conductivity distilled water)?
- the authors conclude and state "...marking the first demonstration of such an operando approach in the literature."* I believe, while the current paper goes far and beyond any others in terms of spatial and temporal resolution and individual electrode detection, there are other demonstrations of "operando battery visualization via ultrasound" in the literature (e.g. <https://pubs.acs.org/doi/full/10.1021/acsenerylett.1c01324>)

Reviewer #2

(Remarks to the Author)

The manuscript discusses a method to obtain waveforms via scanning acoustic microscopy (SAM) and then perform signal analysis to determine heterogeneous features in a cycling lithium-ion battery (figure 1), particularly lithium metal deposition. The image fidelity is demonstrated in Figure 2 over time, with examples of feature analysis as a function of electrochemical signature. reference electrode analysis and post-mortem images (figures 3 and 4) corroborate the growth of Li metal upon the graphite, and the SEM-EDX confirms lithium-containing growths (though the morphology more than the EDX results indicate lithium metal deposition).

The standout feature of this effort is the combination of temperature and spatial resolution of the resulting images. It is an effective distillation and application of previous literature in the space (as cited by the effort), and the method is well described. I have only a few comments and questions.

- What is the impact of poor electrolyte fill on the resultant signal, and how does “electrolyte lean” loading impact lithium plating? It is my understanding that it exacerbates this, but the lack of electrolytes may also preclude the electrochemical signals. Were other electrolyte loading tested?
- Were any “higher rate” tests done? I’m curious how this method works with a > 3 C charging protocol.
- Can different morphologies be classified and identified by this method? There is increasing interest in differentiating mossy from compact growth.

Version 1:

Reviewer comments:

Reviewer #1

(Remarks to the Author)

I reiterate my prior positive review of the work and emphasize its distinct ability to image the dynamics of local Li plating. While prior work on Li plating detection using ultrasound has been done, the presented work is of substantial progress because it indicates the ability to detect local Li plating by focusing their transducer at various depths in the cell with substantial improvement in spatial resolution. The authors have also done a great job in validating their methodology and hypotheses. Their additional responses provide further clarification on the specifics of the work, and instill confidence in the ability to get depth resolution within closed form pouch cells through transducer focusing and pulse-echo gating.

We thank both reviewers and the editorial staff for their time and effort to improve our manuscript. We addressed all points individually, following each comment.

Reviewer #1 (Remarks to the Author):

This review is for the submitted paper “Operando Visualization of Lithium Plating by Ultrasound Imaging of Battery Cells” by Wasylowski et al. While development of ultrasound for non-destructive characterization of batteries has been well substantiated in prior studies, I believe the current work is of potential significant novelty for the field and I recommend publication in Nature Communications pending the revisions as described below. One of the key critiques of ultrasound, in comparison to neutron imaging or X-rays, in the detection of Li plating for fast charging is the lack of spatial resolution, and in particular, the inability to resolve degradation within an individual electrode layer for a multilayered closed form battery. While one might argue that exact localization of Li plating is not necessarily needed, since gauging SOC/SOH may be sufficient in determining a bad cell, pushing the boundaries of what ultrasound can achieve is still an important research need. This paper begins to show that local lithium deposition, not just spatially across a battery, but also within the depth of the cell stack, can be achieved with sufficient signal processing and advanced scanning acoustic microscopy hardware.

- p. 5: the authors state that the brightness becomes higher during charge. My understanding is that this brightness is the intensity of the amplitude that is reflected back. However, previous studies using transmission mode indicate an increase in transmission amplitude upon charging. Should the pulse echo changes be opposite that of the transmission mode? (waves that transmit better should have poorer echo intensity).

We thank the reviewer for the feedback and this comment. The concept of analyzing both reflection and transmission rarely gets attention in the literature and sparks an interesting discussion. While the small increase in brightness between state (1) and (2) in Fig. 2a is not the focus of this work, we agree that it is not intuitively understandable based on existing ultrasound transmission literature. Therefore, we discuss a wide range of possibilities that might explain our observation:

We agree with the finding that the transmission amplitude generally rises upon charging (examples: 10.1149/2.1411712jes, 10.1039/C5EE00111K, 10.1016/j.jpowsour.2017.01.090). Based on the examples by Hsieh et al., Davies et al. and Gold et al., this value is around 2-4% for commercial lithium-ion cells with a graphite anode. It is also correct that if we solely rely on the assumption that the reflected wave is the difference between the emitted and transmitted waves, then the changes in the emitted and transmitted waves should be opposite to each other. However, when accounting for wave attenuation losses (diffraction, scattering and absorption) within the specimen this is no longer necessarily the case. As shown in the publication by Hsieh et al., the reflected and transmitted wave amplitude changes show a highly similar trend (10.1039/C5EE00111K Fig. 3 for a prismatic cell). This observation is consistent with our measurement from Fig. 2a (increase of amplitude upon charging). Hsieh et al. draw the conclusion that changes in the transmission amplitude are mainly caused by the changes in attenuation rather than solely increased reflection. Therefore, a more general equation for the transmission intensity/amplitude would be

$$A_t = A_i - A_r - A_l,$$

with A_l denoting the amplitude losses, A_t the transmitted amplitude, A_r the reflected amplitude, and A_i the injected amplitude. The quantification of ultrasound wave attenuation depending on wave frequency and specimen parameters (coating thickness, current collector thickness, electrolyte volume, ...) remains an open field of research and shall not be the focus of this work (see 10.1002/ente.202200861).

Based on the similarity of reflected and transmitted wave amplitude changes presented by Hsieh et al. we are convinced that the decrease in attenuation is the dominating effect that leads to the increase in reflection amplitude upon charging.

To keep the focus of the manuscript on plating detection, we added a summary of this discussion in the new chapter “Explanation regarding the increase in reflection amplitude upon charging” to the supplementary material. We additionally added a reference on page 6 of the main manuscript: Based on the findings of Hsieh et al. and Davies et al., it can be assumed that changes in attenuation have a highly similar effect on the transmission and reflection amplitude during cycling [30, 31]. A detailed explanation of the increase in reflection amplitude between states (1) and (2) can be found in the supplementary material.

- on the other hand, the authors’ explanation that the bright white spots (p. 6) indicate Li plating appears to make sense, as thick Li deposits are likely to induce transmission amplitude attenuation, and thus greater wave reflections. is this correct?

It is correct that an additional Li-metal layer results in a stronger reflection and thus a reduction in the transmitted amplitude. In contrast to the first comment, the attenuation of the ultrasonic waves by the remaining cell components (caused by diffraction, scattering and absorption) is assumed much smaller than the reflection caused by impedance mismatch during Li-plating. Therefore, there should be an increase in the reflected amplitude and a decrease in the transmitted amplitude. This aligns with the literature, for example the publication by Bommier et al. (10.1016/j.xcrp.2020.100035). To make this clear in the manuscript, we added the following lines on page 6: **The transmitted amplitude is assumed to be decreased because of the increased reflection caused by the Li deposits [29]**

- were the adhesives placed on ALL anode sheets, or just the anode sheet closest to the transducer?

The adhesives were manually placed on all anode sheets in a semi-alternating pattern displayed in the additionally inserted supplementary Fig. 6. The alternating positions of the adhesive dots are intended to prevent severe local geometric inhomogeneity, which would occur if two adhesive dots were placed directly on top of each other. Additionally, the outer-most layers include extra adhesive in all four corners of the anode sheet to enhance mechanical stability during the assembly process. To make this clear, we have added an “Adhesive pattern” section in the supplementary material:

The PVDF and acetone-based adhesive dots were placed in a semi-alternating pattern to ensure a homogeneous electrode stack despite local inhomogeneities. These adhesive dots, as shown in supplementary Fig. 5, were applied only to the anodes and not to the cathodes. While their primary role in this work is to trigger lithium plating, these adhesives are typically used to enhance bonding between the electrode and separator in commercial battery cell manufacturing [4]. This helps in aligning the battery cell components during assembly. Based on this, additional adhesive dots were added to the four corners of the outermost anodes.

Additionally, we reference the adhesive pattern in the main manuscript on page 4: **The adhesive pattern throughout the electrode stack can be viewed in supplementary Fig. 5**

Supplementary Fig. 5: Adhesive placement pattern on the anode sheets. Alternating the positioning of the adhesive dots leads to a more homogeneous geometric shape of the electrode stack, as this prevents two adhesive dots from lying on top of each other.

- Fig 4 the authors depict the optical image of the anode sheet that is closest to the focusing length of the transducer. What is this length and which anode sheet is this? I think it would be more clear if the authors put a few acoustic images next to the same optical images for direct comparison.

Thank you for pointing out this need for more clarity. As mentioned on pages 4 and 5, all layers analyzed so far are the first anode layers in the cells (after the first two separator layers and the first cathode layer). We chose this approach because, for this initial demonstration, minimizing the attenuation of the sound wave by the specimen would result in the highest quality images. Here, the focusing length of the transducer is effectively constant at 1.905 cm. The transducer is positioned according to chapter 2.1 so that it is about 1.905 cm away from the targeted layer. To visualize the first anode layer we position the transducer 1.885 cm away from the pouch cell housing (as mentioned on page 5). To clarify the focused anode sheet directly in context of the figure, **we have adjusted the caption of Fig. 4.**

To ensure a direct comparison between the ultrasound images and optical images, **we have added both optical and corresponding ultrasound images to Fig. 4.** By separately analyzing different gates of the ultrasonic signal, we can visualize different deposits on both sides of the electrode. Therefore, the images of both sides of the anodes have been transparently superimposed. Accordingly, the corresponding ultrasound images (analysis of gate 3 and gate 4) have also been transparently superimposed. **These are presented in the adapted Fig. 4 c and d in the manuscript.**

It is important to note that while the direct comparison shows many similarities between the ultrasound and optical images, the post-mortem optical images do not necessarily represent the real-time state of the anodes, as the cell was disassembled 10-20 minutes after the last ultrasound image was taken. Despite this, the correspondence between the ultrasound images and the optical images is noteworthy.

We reference this in the manuscript with the following lines: **In Figure 4c and d, we present a direct comparison of the optical and ultrasound images of cells 1 and 2. To achieve this, we superimposed the optical images of both sides of the first layer anode in Figure 4c, along with the resulting ultrasound images based on the analysis of both gates 3 and 4 in Figure 4d. Despite the optical images being taken post-mortem, this correlation between both imaging methods remains noteworthy.**

Fig. 4 Optical images of the first anode layer visualized by ultrasound. These images were extracted during the post-mortem analysis of cell 1 at a cell voltage of 4.2 V. **a** shows the front side of the anode sheet. Markings (1) and (3)-(6) indicate silver depositions near or on top of the adhesive dots. Marking (2) indicates a gray deposition in the center of the anode sheet, which is partially congruent with the Y-shaped deposition on the backside of the same anode sheet in **b** marked as (9). Marking (7) tags a reference area of the anode. Markings (8) and (8*) show a large agglomeration of silver depositions near the cathode tab and an adjacent adhesive dot. These silver depositions correspond to the bright indications seen in the ultrasound images. **c** and **d** illustrate optical and ultrasound images of cell 1 and 2 for direct comparison. For the optical images in **c**, we transparently overlaid images from the front and back side of the first anode layer. For the ultrasound images in **d**, we transparently overlaid the images generated by analyzing gate 3 and gate 4.

Further, for a complete demonstration, the authors should then change the focus length and show another comparison of another electrode layer further away from the transmitting transducer. is this possible, or is it only possible to look at the first few layers? This part is particularly important to address. I imagine a 25 MHz wave cannot penetrate too deeply. What does it mean if the technique can only see the first few layers - do we need lower frequency transducers? Along the same lines, these frequencies are still longer wavelengths than individual electrode layers (and therefore, waves could overlap between layers and/or exhibit complicated partial wave reflections).

Thank you for this helpful comment. Even though the focus of this paper is on demonstrating the operando detectability of Li-plating and the chosen sensor is by no means optimal for every cell and research objective, we agree that the visualization of phenomena on different layers preferably within a whole battery cell is of great interest. Certainly, at a sound frequency of 25 MHz, a significantly lower penetration depth is to be expected than at lower frequencies. According to the studies on the speed of sound within cells by Gold et al. and Feiler et al. (speed of sound c approx. 1650 m/s) and using the equation $\lambda = c/f$, the wavelength corresponds to

$$\lambda_{25\text{ MHz}} = 1650\text{ m/s} / 25\text{ MHz} = 66\ \mu\text{m}.$$

This wavelength would become longer at lower frequencies, resulting in potentially lower resolution but higher sample penetration. Although we are by no means convinced that the chosen sensor is the optimal one for use on thick battery cells, we would still like to demonstrate its usefulness for more than the first few cell layers. Since no pristine cell from Fraunhofer ISIT was available after the preliminary studies and the main study in the manuscript, as well as the verification with a third

electrode, we have chosen a common alternative cell from the manufacturer LiFun (model number 5575166) for this demonstration.

For a clear demonstration of the detection of effects through all electrode layers of the cell (9 anodes, 8 cathodes double-sided coated), a circular indentation was carefully pressed onto the back of the cell and visualized from the front by lowering the sensor to the cell by the thickness of the cell. Although visualization of the indentation is possible, the image of the back of the cell is quite inhomogeneous, which can be explained by the weakness of the reflection at the focal point. Reflections outside the focal point therefore also start to be of the same order of magnitude as the deep reflection in the focal point. This disadvantage could be overcome by a lower sound frequency or stronger focusing (smaller focal zone). A representative sensor study can therefore be the focus of future work. **The test description and the results of the examination are shown in the new supplementary Fig. 6 and in the supplementary material in the new chapter "Possible ultrasound detection depth".**

Supplementary Fig. 6: Demonstration of the possible penetration depth using 25 MHz Transducers. a shows optical images of the sample LiFun 5575166. b shows the corresponding ultrasound images of the sample at different sensor distances, which produces images at different depths within the cell. It is possible to visualize mechanical features on the other side of the cell with the same 25 MHz sensors. However, the image sharpness at greater depths could be increased by using lower sound frequencies.

We reference this discussion in the main manuscript on page 4 with the following lines: “While deeper layers can be visualized, image quality may be inferior when using 25 MHz transducers compared to lower frequency transducers (see supplementary Fig. 6 for a comparison of visualization at the first and last layers of a 4 mm thick cell).”

- how do the authors guarantee there is absolutely no gas formation or electrolyte dry out in their cells? how does one decouple these failure modes over longer cycles? e.g.

[https://www.cell.com/joule/pdf/S2542-4351\(23\)00440-3.pdf](https://www.cell.com/joule/pdf/S2542-4351(23)00440-3.pdf) demonstrated that spatial heterogeneities were due to electrolyte consumption

This point is well taken, especially since many publications on ultrasound imaging of battery cells deal with electrolyte filling, dry out or gas detection. Nevertheless, for the following reasons, we are convinced that no other failure modes significantly influence our measurement:

Electrolyte dry out: Since we see a contraction of the bright spots in the ultrasound images at the beginning of the CV phase, during relaxation after charging and when the cell is being discharged, and a ring-like expansion upon subsequent charging, and electrolyte dry out is considered a non-reversible process, it can be assumed that this effect did not play a significant role in our investigations.

Gas formation: Similar to electrolyte dry out, gas generation is also a non-reversible process in most cases. However, it could be argued that if the current is reduced, the gas generation could stop, and the gas could partially migrate into cavities that are not on the surface of the electrode. To have a direct comparison to ultrasound images with provoked gas generation, calendar aging tests were carried out in a separate study. Here, gas generation was specifically provoked in a Fraunhofer cell. The corresponding ultrasound image can be seen in **supplementary Fig. 7 a**. In contrast to the ultrasound images in Fig. 2 of the manuscript, the gas regions in the ultrasound image of **supplementary Fig. 7 a** are smooth and homogeneous, making it impossible to detect the electrode structure. Additionally, gas would need to be generated precisely on the lithium deposits and migrate laterally, as it would remain in the focal zone during vertical migration into the electrode. An argument against this occurring in the cells shown in Fig. 2 is that ultrasound images of cell 2, taken over several hours after charging without electrochemical cycling, show no migration to or from the bright spots.

In a separate study, a different pouch cell was not fully degassed. Due to incorrectly set parameters in the vacuum regulator and an insufficiently deep-drawn pouch casing, only the gas in the center of the cell surface was removed, leaving some gas at the edge of the cell. The corresponding ultrasound image with a different color map can be seen in **supplementary Fig. 7 b**. It is again clearly recognizable that there is no information about the structural details of the electrodes in the gas areas and that there is a sharp edge to the non-gassed areas. Once again, the ultrasound image at the gas points is smooth and homogeneous in contrast to the spots in Fig. 2.

Supplementary Fig. 7: Ultrasound images of gassing in different pouch cells. The images were created using the same settings and transducer as described in section 4.3. **a** shows the ultrasound image of a cell in which gas generation has been provoked by calendar aging with the gas accumulating in the center of the cell surface. **b** shows another cell in which gas has accumulated at the edge of the cell due to failed degassing and insufficient deep drawing of the housing. In both cases, the gas is outlined by a clear line, causes a homogeneous reflection and does not allow interpretation of the electrode structure under the gas.

In summary, based on the visual differences in the gas accumulation ultrasound images from supplementary Fig. 7 and the indications from the manuscript (Fig. 2), as well as the decreasing and increasing size of the indications upon cycling without observed gas migration, we do not assume that local gas generation took place exactly above the Li deposits in this study. Alternatively, to the verification steps in the revised manuscript such ultrasound examinations could be performed in parallel to operando neutron imaging in future studies to obtain additional verification of the results during the initial stages of the ultrasound imaging method (10.1038/srep15627).

We reference supplementary Fig 7 and this discussion on page 12 in the Verification chapter of the manuscript: “An increased ultrasound reflection, which would result in a brighter pixel when analyzing gate 1 instead of gate 3, could, in principle, also be caused by local gas generation. However, a comparison of ultrasound images with the same sensor selection and the same measurement settings shows a different ultrasound image at gassed locations of other comparable cells (see supplementary Fig. 7). Due to the partially reversible size and shape of the indications in correlation with the electrochemical data, it is also not assumed that the indications represent another failure mode such as electrolyte dry out [41].”

Additionally, we added a section called “**Alternative failure modes**” to the supplementary material which explains in more detail, why we are not convinced that electrolyte dry out or gas formation is causing the indications in the ultrasound image.

How failure modes in ultrasound images can be quantitatively distinguished based on signal properties is still subject of current research. Typically, gas accumulations would not allow penetration into deeper layers (even of the focal zone), resulting in the smooth homogeneous surfaces in the ultrasound image. In addition, gas generation usually leads to a local expansion of the cell, which could be detected via the time of flight of the reflected signal. As the lack of electrolyte leads to increased attenuation of the ultrasound wave, a possible approach to detecting electrolyte dry out would be to detect if only the structure of the first few layers can be imaged, in contrast to locations with more electrolyte.

For the quantitative detection of deposits, the expected reflection coefficient in the focus zone could be calculated in a similar way to publication 10.1016/j.jpowsour.2023.233295 and used to deduce a material transition. However, this is a very challenging endeavor, as it requires very precise positioning of an ideally very small focus zone within the cell, minimal deviations in the coating thickness, and a generally very homogeneous structure. A further drawback of this method is that many deposits in the battery could have similar acoustic impedances and therefore require very precise measurement electronics to distinguish between them.

- For the SEM/EDS, were the samples thoroughly rinsed in solvent and dried before imaging?

Regarding the statement about reaction with LiPF₆ - I'm not sure if this is a common belief in battery literature, that the dissolved LiPF₆ reacts (without any applied current/potential). Solvent reduction should be the major degradation mode.

Thank you for this valid comment. The samples were not rinsed, but punched out directly after cell disassembly and photo scanning and analyzed in the SEM/EDS. The purpose of this procedure is that we may analyze the electrodes close to how they were present in the cell. Rinsing could remove things of potential interest. The reviewer's comment regarding the LiPF₆ reduction as a strong aging phenomenon is well taken. In response, we have revised our hypothesis from the supplementary material: Upon closer analysis of the EDS spectrum of sample (9) in Fig. 5 of the manuscript, a fluorine-heavy F:P (14.5:1) ratio is noticeable. This indicates that LiPF₆ cannot be the only fluorine-containing deposit at this location for stoichiometric reasons. However, due to the Y-shaped structure of the deposits in the center of the cell, we still hypothesize that the deposits are indirectly related to vacuum drawing and the adhesive dots. This is also supported by the fluorine-containing adhesive (PVDF and acetone mixture). Due to the obviously different condition of the adhesive in cell 1, we assume that either small parts of the adhesive itself (in the liquid state) or components of it (after reaction with electrolytes) were moved into the center of the cell in a Y-shape from the adhesive points during vacuum drawing. In the case of the adhesive (and not its reaction products), however, the

quantities would have to be so small that there was apparently no influence on the Li-ion flux density at these points.

We have thus adapted large parts of the chapter "**Explanation regarding the gray deposits in the center of cell 1**" of the supplementary material based on the above discussion.

- In the conclusion the authors state the hardware is low cost as a benefit. The Sonoscan is at least \$200K USD - is this still "low cost" or "enabling widespread use and scalability"? Would any battery manufacturers be willing to submerge their large cells in water (even though its low conductivity distilled water)?

We agree that the device used in this study is currently offered at a price that is not suitable for the mass market. In contrast to the current price for such a device, the minimum necessary hardware costs for ultrasound imaging are still low. This was demonstrated by the open-hardware gantry setup in our earlier publication (10.1016/j.jpowsour.2021.230825) or other inexpensive setups such as those by Chang et al. (10.1021/acseenergylett.1c01324). The device used in our study is a laboratory device specifically designed for defect detection in microelectronics and offers many options that are not essential for battery analysis. Although quantitative price development is not the focus of this work it is certainly feasible that successful demonstrations of the measurement methodology will result in devices being offered on the market by other manufacturers who offer cheaper alternative products (with adjusted features) in potentially higher quantities. Thus, to complement the research of the measurement method's potential, future work should continue to include demonstrations of its cost-effective implementation and its scalability and integration into development processes.

Regarding the willingness of cell developers or manufacturers to immerse large cells in water for testing, our institute has had many positive experiences, which unfortunately must remain confidential. If water immersion is not preferred, there are several alternatives available:

- Water jet method: Both in the Sonoscan and in an inexpensive, self-built version at our institute, a jet of water can be directed in front of the sensor with a pump and immediately suctioned away at the cell. This method ensures that the cell only encounters water at the sensor location.
- Alternative coupling agents: Many publications have reported the successful use of mineral oil or silicone oil as coupling agents for ultrasound imaging. These also work in combination with the jet method.
- Air-coupled ultrasound: This method allows the use of ultrasound without a liquid coupling agent, though it is generally limited to frequencies below 1 MHz.
- Laser-induced ultrasound: For using ultrasound in the MHz range without a coupling agent, laser-induced ultrasound can be employed as the emitter with optical microphones as the receiver. This technology is currently being commercialized by companies like Xarion (xarion.com, doi.org/10.1117/12.3000954).

Based on this discussion, we have added the following lines on page 13 in the "Conclusion" chapter.

"... This significantly reduces the barrier to entry for spatially resolved operando aging studies and enables parallelization in cell development due to the **low cost of the ultrasound imaging hardware requirements**. While current commercial ultrasound devices are offered at a high price, the focus of this work is on demonstrating the potential of the measurement method. Work on the low-cost implementation of the measurement method can thus help to promote widespread implementation of the technology."

With regard to alternative coupling agents, we have added the following lines on page 13 in the "Conclusion" chapter.

"Alternatively, based on the introduction of a water jet in front of the sensor and direct suction at the cell, it can be ensured that the specimen only comes into contact with liquids at the sensor location. If the use of water is to be avoided, alternative liquids such as mineral or silicone oil can be used. Alternatively, laser-induced ultrasound can be used to avoid the use of liquid altogether [43]."

- the authors conclude and state "...marking the first demonstration of such an operando approach in the literature."* I believe, while the current paper goes far and beyond any others in terms of spatial and temporal resolution and individual electrode detection, there are other demonstrations of "operando battery visualization via ultrasound" in the literature (e.g. <https://pubs.acs.org/doi/full/10.1021/acsenergylett.1c01324>)

We thank the reviewer for this comment and acknowledge the pioneering work of, for example, Chang et al. or Robinson et al. (10.1021/acsenergylett.1c01324, 10.1039/C8CP07098A, 10.1149/1945-7111/abb174). In order to recognize the significance of the previous work, we have made the following changes to the manuscript:

- We deleted the line referenced by the reviewer "... marking the first demonstration of such an operando approach in the literature." and instead write: "... demonstrating a significant advancement of this operando method."
- In addition, we reference and cite the previous work of Chang et al. and Robinson et al. in the introduction on page 3: "Contrary to the pioneering transmission-based work by Chang et al. and Robinson et al., this method is based on the emission and reflection of ultrasound waves, ..."

Reviewer #2 (Remarks to the Author):

The manuscript discusses a method to obtain waveforms via scanning acoustic microscopy (SAM) and then perform signal analysis to determine heterogeneous features in a cycling lithium-ion battery (figure 1), particularly lithium metal deposition. The image fidelity is demonstrated in Figure 2 over time, with examples of feature analysis as a function of electrochemical signature. reference electrode analysis and post-mortem images (figures 3 and 4) corroborate the growth of Li metal upon the graphite, and the SEM-EDX confirms lithium-containing growths (though the morphology more than the EDX results indicate lithium metal deposition). The standout feature of this effort is the combination of temperature and spatial resolution of the resulting images. It is an effective distillation and application of previous literature in the space (as cited by the effort), and the method is well described. I have only a few comments and questions.

- What is the impact of poor electrolyte fill on the resultant signal, and how does "electrolyte lean" loading impact lithium plating? It is my understanding that it exacerbates this, but the lack of electrolytes may also preclude the electrochemical signals. Were other electrolyte loading tested?

Thank you for the feedback and this interesting question. Insufficient electrolyte volumes have already been investigated in previous studies on ultrasound imaging of battery cells. In a study by Deng et al. (10.1016/j.joule.2020.07.014), the amount of electrolyte was first varied (between 0.4 mL and 0.9 mL for a 240 mAh NMC532/AG pouch cell) without applying potential (open-circuit conditions). Since electrolyte strongly reduces the attenuation of ultrasound propagation, missing electrolyte can be detected by a strong decrease in ultrasound amplitude in ultrasound transmission. However, since we are investigating ultrasonic reflection instead of transmission, this investigation is more complicated. For this purpose, the penetration of the sound wave into deeper electrode layers must be monitored, as already commented in Reviewer 1. This can be done by precisely positioning the focus point of the sensor in deeper ultrasonic layers and/or by analyzing the corresponding ultrasonic signal.

If there is insufficient electrolyte in the cell, an inhomogeneity should be visible in the ultrasound image even before charging or applying potential. With subsequent cycling, as already indicated by the reviewer, it is likely that the inhomogeneities in the ultrasound image will worsen. This was also addressed in the publication by Deng et al. in the supplementary material (Figure S2, 10.1016/j.joule.2020.07.014). Here it can be seen that before cycling, the poorly wetted cells already have a strongly attenuated ultrasound signal in the middle of the cell. During cycling, the ultrasonic signal in the middle of the cell drops to zero, which usually indicates gas generation, as ultrasound cannot propagate through gas. If, in our case, lithium plating was also provoked on poorly wetted cells,

the reflection of the ultrasound would increase over time due to the reduced transmission (brighter pixels with increasing thickness of the lithium plating).

However, as the focus of this work is only on demonstrating the detectability of lithium plating, future work should investigate and demonstrate the limits of this investigation method for combinations of failure modes (e.g. electrolyte deficiency and Li plating) and develop methods for quantitatively distinguishing these effects. This could be done, for example, by using a combination of sensors and excitation methods for different detection methods. For example, transmission measurements are particularly sensitive to gas formation, but less sensitive to effects on individual electrodes. To motivate this research question in the manuscript, we have adapted the following lines in the conclusion on page 13: “Since we have actively tried to circumvent other aging effects in this experiment, a further open research question is how the presented method could be used to quantitatively distinguish between overlapping aging mechanisms. For example, a combination of transmission and reflection measurements would be feasible to exploit different sensitivities of the modes to other aging effects.”

- Were any “higher rate” tests done? I’m curious how this method works with a > 3 C charging protocol.

Since the selected current (1 C) for these tests already matched the maximum current for this cell according to the manufacturer's specifications and was sufficient to provoke Li plating in our cell, no higher currents were applied. However, higher currents would potentially cause an increased temperature gradient, apart from increased aging effects in our cells. This gradient is usually increased due to inhomogeneous current density distribution. Since, according to publications such as those by Owen et al. (10.1149/1945-7111/ac6833), there is a linear decrease in ultrasound amplitude with respect to temperature, this would also be visible implicitly in the image. In a study by Waldmann et al. (10.1149/2.0561506jes), a temperature gradient of about 4 K was detected at a 3 C charge rate in large-format pouch cells. Based on the results of Owen et al. at an ambient temperature of 25 °C, this would correspond to an amplitude reduction of about 3%. Since this amplitude change is below the lithium plating-related changes, this effect does not significantly limit the method presented in this paper. However, this temperature gradient would, for example, make local SoC determination more difficult, since the SoC-induced amplitude change is of the same order of magnitude.

Thus, we have added the following outlook to the conclusion chapter on page 13 in the manuscript: **Furthermore, the compensation of SoC- or temperature-induced amplitude changes through cycling at high current rates should be considered in future work, as this can result in amplitude errors of up to 3% [44,45].**

- Can different morphologies be classified and identified by this method? There is increasing interest in differentiating mossy from compact growth.

We agree with the reviewer's observation that the distinction between different lithium plating morphologies is of great interest. For example, in the publication by Kühnle et al. up to 4 main categories (and many more subcategories) of plating morphologies could be identified. In order to distinguish the subtle differences in local plating occurrences, one would need a sub-micrometer resolution comparable to electron microscopes. On the basis of empirical preliminary investigations, no improvement of the ultrasound image below pixel widths of 75 μm could be found for a large number of sensors (see Fig. R2). This is mainly due to the maximum focusability of the ultrasonic

Fig. R2: Illustration of the focusing of ultrasonic beams with 6mm active element transducers. For a resolution of smaller structures such as platinum morphologies, the focusing of the sensors must be greatly enhanced. The size of the sensor's beam and the focus point are not shown to scale and are for illustrative purposes only.

beam of current piezoceramics. It can therefore be assumed that the morphology of individual lithium deposits cannot currently be visualized using ultrasound imaging. For this purpose, special piezoceramics or acoustic lenses would have to be manufactured, which have a corresponding degree of diffraction to achieve a higher resolution. However, it would be plausible to examine the macroscopic structure of the plating more closely in future work. For example, solid&smooth plating could possibly be distinguished from fiborous&rough plating. For this purpose, very high-resolution sensors could be applied to a structure with ideally a single-layer cell in which different plating forms are provoked in a comparable manner (see publication by Kühnle et al.). However, transferability to larger cells would be more difficult due to the increased attenuation of high-resolution ultrasonic waves.

Based on the above discussion, we have included the following line in the conclusion of the manuscript on page 13: "Another potential area of application of the presented method using higher resolution ultrasonic sensors would be the real-time investigation of macroscopic plating morphologies as proposed by Kühnle et al [37]."